# Electrical Conductivity of Additively Manufactured Copper and Silver for Electrical Winding Applications

**DOI:** 10.3390/ma15217563

**Published:** 2022-10-28

**Authors:** John Robinson, Sai Priya Munagala, Arun Arjunan, Nick Simpson, Ryan Jones, Ahmad Baroutaji, Loganathan T. Govindaraman, Iain Lyall

**Affiliations:** 1Additive Manufacturing of Functional Materials (AMFM) Research Group, School of Engineering, University of Wolverhampton, Telford Innovation Campus, Telford TF2 9NT, UK; 2Additive Analytics Ltd., Telford TF3 1EB, UK; 3Aceon Group, Telford TF3 3BJ, UK; 4Electrical Energy Management Group, Department of Electrical and Electronic Engineering, University of Bristol, Bristol BS8 1TR, UK; 5Anopol Ltd., Birmingham B5 5QA, UK

**Keywords:** additive manufacturing, 3D printing, laser powder bed fusion, copper, silver, electrical resistivity, electrical conductivity

## Abstract

Efficient and power-dense electrical machines are critical in driving the next generation of green energy technologies for many industries including automotive, aerospace and energy. However, one of the primary requirements to enable this is the fabrication of compact custom windings with optimised materials and geometries. Electrical machine windings rely on highly electrically conductive materials, and therefore, the Additive Manufacturing (AM) of custom copper (Cu) and silver (Ag) windings offers opportunities to simultaneously improve efficiency through optimised materials, custom geometries and topology and thermal management through integrated cooling strategies. Laser Powder Bed Fusion (L-PBF) is the most mature AM technology for metals, however, laser processing highly reflective and conductive metals such as Cu and Ag is highly challenging due to insufficient energy absorption. In this regard, this study details the 400 W L-PBF processing of high-purity Cu, Ag and Cu–Ag alloys and the resultant electrical conductivity performance. Six Cu and Ag material variants are investigated in four comparative studies characterising the influence of material composition, powder recoating, laser exposure and electropolishing. The highest density and electrical conductivity achieved was 88% and 73% IACS, respectively. To aid in the application of electrical insulation coatings, electropolishing parameters are established to improve surface roughness. Finally, proof-of-concept electrical machine coils are fabricated, highlighting the potential for 400 W L-PBF processing of Cu and Ag, extending the current state of the art.

## 1. Introduction

High-efficiency and high-power-density electrical machines will play a vital role in decarbonisation efforts and in meeting Net Zero Emission targets across the automotive, aerospace and energy sectors. Electrical machine technology roadmaps published by the Advanced Propulsion Centre (APC) [1] and Aerospace Technology Institute (ATI) [2] seek continuous power density of between 9 and 25 kW/kg by 2035. These targets are in stark contrast to the 2–5 kW/kg available at present, representing a significant challenge [1,3]. Electrical machine windings rely on highly electrically conductive materials such as copper (Cu) and aluminium (Al) and, while silver (Ag) exhibits higher electrical conductivity, it is typically prohibitive by its relatively high cost [4]. Step changes in electrical machine power density could be realised by simultaneously improving efficiency through optimised materials, targeted conductor geometry and topology and integrated thermal management strategies. In this regard, Additive Manufacturing (AM) offers unparalleled design freedom, enabling optimum winding geometries that are not feasible through traditional manufacturing techniques [5]. For AM of metals, the Laser Powder Bed Fusion (L-PBF) process is the most mature and advanced technology and has been widely adopted to fabricate medical, aerospace and automotive metal components [6,7]. During L-PBF AM, components are fabricated in a layer-by-layer process through the Selective Laser Melting (SLM) of fine metal powder layers on precedingly melted layers. Therefore, the L-PBF process is highly dependent on the laser and material interaction, which is dictated by the associated laser parameters selected [8]. Some of the main variables include the metal powder feedstock composition, powder Particle Size Distribution (PSD), laser power, laser scan speed, laser spot size and layer thickness. These in turn effect the phase transformation of the metal, molten metal flow and cooling, laser diffusion, scattering, absorptivity and reflection [9].

Additionally, laser wavelengths in the infrared range (1060–1090 nm) are commonly used in L-PBF systems due to favourable absorption for common L-PBF metals, such as steels that reflect only ~40% of thermal energy [10]. However, solid Cu commonly featured in electrical coil windings can reflect up to ~98% of the laser energy at infrared laser wavelengths [11] and, in powder form, Cu still reflects ~71% of laser energy [10]. Furthermore, feedstock PSD can affect powder spreading, packing density and laser powder interaction [12,13,14], and laser powder interactions can vary with geometry and height in Z [8,15]. Therefore, an understanding of the material–laser interaction to maximise material absorptivity while limiting laser reflectivity is fundamental [9,16] for successful L-PBF processing of Cu and Ag [17,18]. Previous approaches to mitigate the challenges of L-PBF processing Cu include surface modification of powders [19], addition of nanoparticular substrates [20], alloying [21,22], using alternative wavelength lasers [23,24], higher power lasers and modified scan speeds [19,25]. Colopi et al. [25] have successfully processed high-purity Cu with a final density of ~98% using high-powered (>600 W) infrared lasers. Gargalis et al. [26] compared the laser energy absorptivity between solid and powder Cu powder (99.9%). It was reported that the energy absorptivity of a 100 µm Cu layer is four times that of a solid Cu sample due to the lower thermal conductivity and less reflective nature of the Cu powder feedstock. However, the research also featured a relatively high laser power (>500 W) and slow scan speeds (~150 mm/s), indicating the requirement for a high energy density. Contrastingly, Silbernagel et al. [27] used a relatively low-powered laser (200 W) and small laser spot L-PBF system to successfully assess the L-PBF processing of high-purity (99.9%) Cu for electrical applications. Although the maximum density achieved ranged between 80.3–85.8% (dependent on scan strategy), the relatively porous L-PBF Cu had higher electrical conductivity than L-PBF aluminium alloy AlSi10Mg; achieving 54% International Annealed Copper Standard (IACS) in comparison with AlSi10Mg, which achieved 29% IACS [27]. 

Adding to the challenge, Cu’s oxidisation tendency can result in the formation of oxide layers on powder feedstock during storage and L-PBF processing [28,29]. As such, inert gas powder storage and L-PBF build environment control, such as custom gas mixtures and reducing the L-PBF process oxygen content also offer potential for improved laser processing and component fabrication. However, currently, studies investigating gas mixes and oxygen reduction have primarily focused on L-PBF processing of titanium (Ti) [30,31,32], while investigations of oxygen content for processing Cu have largely been focused on electron beam processing, with little investigation regarding L-PBF [33,34]. 

Due to the increased demand for AM of Cu, the AM industry has developed different variants of Cu for specific applications such as heat exchangers, induction heater coils and electrical windings that offer improved electrical and thermal conductive performance. These variants include high-purity and alloyed Cu with different PSDs and associated L-PBF processing parameters [35]. Accordingly, this study compares the electrical performance of three high-purity Cu industry variants, namely ‘Carpenter Additive Cu’, ‘EOS Copper Cu’ and ‘EOS CuCP’. Additionally, these variants are also compared with pure Ag and Cu–Ag alloys fabricated by L-PBF to investigate the resulting electrical conductivity performance. The addition of Ag to pure Cu was intended to improve the structural stability and the corrosion resistance of the alloy without hampering the electrical conductivity of the base metal at possible elevated temperatures. The powders were subjected to particle size and morphology analysis prior to sample fabrication. Once fabricated, the electrical performance was measured, followed by Computer Tomography (CT) analysis. The influence of electropolishing on the surface finish was also studied to assess the potential application of electrical insulation coatings, with the effects of polishing being quantified in terms of surface roughness. Additionally, the effect of the type of powder recoating blade and the number of laser exposures was also observed. Finally, proof-of-concept coil fabrication is demonstrated to ascertain the feasibility of L-PBF processing of coil windings and heat exchangers using a 400 W L-PBF system. This study is therefore the first to comparatively report the L-PBF fabrication of high-purity (>99% purity) Cu, Ag and Cu–Ag alloys and their resulting electrical performance. 

## 2. Material and Methods 

### 2.1. Laser Powder Bed Fusion 

All samples investigated in this study were fabricated using an industrial-grade L-PBF Additive Manufacturing (AM) system on EOS M290 (EOS GmbH, Krailling, Germany). High-purity (>99%) Ag and Cu powders supplied by Legor Spa, Carpenter Additive and EOS GmbH, respectively, were used as feedstock. The EOS M290 L-PBF AM system features a standard 400 W infrared laser system with 100 µm spot size. The SLM process was carried out in an argon atmosphere at an oxygen content below 0.1%. Following fabrication, all samples were removed from the build platforms using non-contact Electrical Discharge Machining (EDM) (EXCETEK, Taichung, Taiwan). 

### 2.2. Powder Characterisation 

Powder morphology and PSD can affect the packing density and flowability of AM powders [36], subsequently dictating the layer packing and material behaviour during the L-PBF process [36,37]. Additionally, the composition and layer packing density can also influence the laser reflectivity and absorptivity at the powder bed [13]. Previous studies investigating the L-PBF processing of high-purity Cu and Ag have reported the enhancement of mechanical and thermal properties through L-PBF in situ Cu–Ag alloying [17,21]. However, the electrical performance of high-purity Cu has seen limited attention, and the electrically conductive properties of L-PBF high-purity Ag and Cu–Ag alloys have yet to be reported. Accordingly, this study considers six feedstock powder compositions as listed in Table 1. Cu, Ag and Cu–Ag feedstock powders, along with their morphology and Particle Volume Distribution (PVD), were characterised using Scanning Electron Microscopy (SEM) and digital imaging particle analysis techniques. SEM analysis was carried out using a Zeiss EVO50 SEM. A Retsch Technology Camsizer X2 was used to validate the particle morphology, size and shape. The Camsizer X2 utilises dynamic image analysis following ISO 13322-2 [38], where a high-resolution optical system consisting of digital cameras evaluates precise particle characteristics at a rate of 300 images per second.

### 2.3. L-PBF Parameters and Sample Manufacture

L-PBF process parameters developed by the authors [8,18] in previous studies (listed in Table 2) were used for initial sample fabrication. L-PBF parameters were kept constant to ensure cause and effect analysis and ensure any resultant variations could be attributed to powder supplier, composition, morphology and PSD.

Considerable research has been conducted regarding L-PBF feedstock [39,40] and its influence on L-PBF process parameters [34,41,42] and, as such, studies are emerging reporting optimised parameters for L-PBF processing of high-purity Cu [19,25,27]. However, the process and parameter development usually focus on increasing energy density at the powder bed to mitigate the reflective nature of Cu. Limited investigations have reported the effects of scanning strategies, laser exposure time and powder dosing on the influence of processing Cu. Furthermore, there is no literature available regarding the laser exposure and powder recoating strategies for highly reflective feedstocks such as Ag and Cu–Ag alloys. As such, this study considers the AM powder dosing methodology and laser exposure strategies to ascertain the effects on L-PBF Cu, Ag and Cu–Ag fabrication. 

### 2.4. Annealing Process

Annealing is a heat treatment process which alters the microstructure of a material to optimise the resultant properties. Heat treatment of high-purity L-PBF Cu, Ag and Cu–Ag alloys have seen limited investigation, leaving a gap in knowledge regarding processing feasibility and resultant material properties, particularly in relation to electrical conductivity. However, relatively high Ag content (>29% Ag) in Cu–Ag alloys fabricated through ingot casting have seen investigation and optimum recrystallisation annealing processes have been reported as 30 min at 500 °C [43]. Subsequent studies have reported this annealing process creates atomic lattice contractions in L-PBF Cu–Ag alloys [21], which when combined with the Cu–Ag interface interactions can increase strength and thermal diffusivity performance [21]. Accordingly, all samples fabricated for this study followed the annealing process. 

### 2.5. Surface Roughness

L-PBF allows the fabrication of complex metallic structures not feasible with more traditional manufacturing techniques. However, to take advantage of the desirable benefits, a key requirement of the process is to fabricate near-net-shape components with limited post-processing, such as support removal or finishing. Support removal is time-consuming and can negatively affect the L-PBF component surface, in turn creating crack initiation sites negatively effecting mechanical performance [44,45]. Additionally, internal supports can be challenging or impossible to remove dependant on the component geometry and feature dimensions. Furthermore, the layer-by-layer nature of the L-PBF process can result in a staircase step effect, resulting in a relatively rough surface [45,46]. While rough surfaces can be desirable for some applications, they are usually undesirable due to crack and failure initiation sites [47] while increasing surface resistance [46]. Surface machining is a useful tool in this respect and is the most common method utilised to improve surface finish. However, machining access can be challenging or impossible for complex L-PBF structures and alternative surface finishing techniques are required.

Electropolishing is an electro-chemical process commonly utilised for reductions in surface roughness at the micron scale. Material is removed using a DC power supply by setting up a cell between the copper component (anodically charged) and a pair of chemically inert electrodes (cathodically charged) facing the largest surfaces of the part within a conductive viscous medium (typically acidic in nature). A thin anodic film forms around the copper anode where transfer of metals ions occurs from the highest profile areas of the surface, where current density is greatest, and film is thinnest. Metal ions are oxidised by the applied DC current and encouraged to dissolve into the surrounding medium. As component surface finish can be critical for parts that require additional processes (such as insulation coatings for electrical windings), the samples were subjected to electropolishing regimes with varying times and current densities to assess the potential for surface roughness improvement. The effects of anodic polishing on L-PBF-manufactured high-purity Cu sample surface roughness and electrical performance are analysed experimentally. Two samples were immersed in 1 litre of a proprietary phosphoric/solvent mix solution containing metal cheating agents (Anopol 30). The high-purity Cu samples were subjected to current densities of 0.4 a/cm^2^ and 0.8 a/cm^2,^ respectively, for a minimum of 10 min using a QPX1200SP rectifier (Aim and Thurlby Thundar Instruments, United Kingdom). The solution was used at ambient temperature with agitation provided by a magnetic stirrer bar at 1000 rpm. Electropolished sample surface roughness was analysed and compared with as-built Cu samples, and surface roughness results are reported using a Keyence VHX-7000 (Keyence Ltd., Milton Keynes, UK) digital microscope.

### 2.6. Density 

Porosity and pore morphology can significantly affect material properties, including mechanical [18] and electrical performance. For L-PBF unoptimised powder feedstock composition, particle distribution and L-PBF process parameters can negatively affect the resultant density, leading to increased porosity content [18]. Industrial grinding and polishing techniques combined with digital microscopy and SEM analysis are common characterisation methods for density and porosity evaluation. Although these methods provide an indication of material density and pore characteristics, the volumetric data in relation to distribution throughout the samples are limited. Therefore, for this study, the materials were analysed through Computed Tomography (CT) to characterise the volumetric porosity and density.

CT is a non-destructive analysis technique suitable for porosity characterisation. However, the CT data outputs will be dependent on the scanning and threshold parameters set by the operator and, as such, the CT results generated are best suited for comparative rather than absolute analysis. Qualitative analysis of the fabricated samples was conducted using micro-CT on Nikon XT H 225ST (Nikon Metrology Inc., Castle Donington, UK) and internal cross-sectional analysis, completed using SEM on Zeiss EVO50 (Carl Zeiss AG, Jena, Germany). Samples were CT processed at a magnification factor of 22, whilst a Sn filter of 0.25 mm thickness was used. A potential of 223 kV with a beam current of 40 µA resulted in a voxel size of 9 µm. Circa 3100 projections were collected from each sample and reconstructed using CT Agent 3D pro software. The resultant images were vertically stacked using VG Studio max 3.4 software. Post-processing for 3D visualisation was conducted using Dragonfly (Object Research Systems Inc. version 2022.1) software. The reconstructed CT data were subjected to porosity measurements using native algorithms in the Dragonfly software. A cylindrical Region of Interest (RoI) of 30 mm^3^ was created in all the samples for a standardised comparative analysis.

### 2.7. Electrical Conductivity 

The electrical conductivity was measured using a small-signal 4-wire Ohm meter (RS Pro RM-805) and via large-signal 4-wire measurement method in the range from 10A to 60A DC (Fluke Norma 4000) in accordance with ASTM B193-16 [48]. The electrical resistivity was measured in two methods having small and big resistance to test the range of conductivity of the additively manufactured samples. The ends of the samples were manually polished to ensure maximum and uniform contact with the testing probes. From the resistance measurements, the conductivity of the samples was calculated. This measurement is mainly influenced by the surface roughness and density of the sample. The presence of peaks and valleys can disrupt the contact area with the testing probes and pores, and any unfused powders are a clear obstruction to the conduction of electricity. 

### 2.8. Design of the Experiments

The L-PBF process is a combination of over 130 feedstocks and process variables related to powder characteristics, process build environment and laser–material interactions [16]. While material feedstock and common L-PBF process parameters such as laser power and scan speed have seen much investigation, process variables including laser exposure duration and powder recoating can also affect the density of the fabricated components. However, these variables have seen limited investigation in the current literature. Therefore, this study also includes the effect of these parameters on the resultant electrical conductivity performance. Due to the numerous variables investigated in this study, four experiments (Figure 1) were conducted considering the effects of material composition, powder recoating and laser exposure on electrical performance and electropolishing effects on surface roughness. Material feedstock characteristics were first ascertained, and SEM and CT were utilised to assess fabricated sample density and porosity distribution. 

Firstly, three high-purity Cu powders were studied to evaluate the influence of purity on the L-PBF processability using a standard 400 W infrared laser. Secondly, high-purity Cu was compared with high-purity Ag and two Cu–Ag alloys (namely Cu–Ag10% and Cu–Ag20%) to identify the influence of Ag addition on sample density and electrical performance. Thirdly, laser exposure and powder recoating methods were investigated to characterise the effects on resultant density and electrical performance. Finally, the effects of electropolishing regimes on high-purity Cu and the effects on electrical performance were investigated before all results were collated and summarised. A total of nine International Electrotechnical Commission (IEC) samples were fabricated, with two samples further electropolished. While additional process parameter development could be utilised for material optimisation, it was outside the scope of this study.

## 3. Results and Discussion 

### 3.1. Powder Feedstock

Powder feedstock morphology and PSD can have a significant impact on L-PBF processability, as they affect the packing density and flowability of the powder bed [39]. Therefore, the Cu, Ag and Cu–Ag powder feedstock morphology, as well as Particle Volume Distribution (PVD), were characterised using Scanning Electron Microscopy (SEM) (Zeiss EVO50) and digital imaging particle analysis techniques. The SEM morphology data for Cu, Ag and Cu–Ag powders are displayed in Figure 2. 

All high-purity Cu and Ag particles feature spherical morphology, which is desirable for L-PBF AM. EOS Cu (Figure 2b,c), Cu–Ag (Figure 2d,e) and Legor Ag (Figure 2f) powders have visibly similar particle size distribution. However, the Carpenter Additive Cu (Figure 2a) powder displayed a visibly lower particle distribution, with a larger proportion of powder particles below 20 µm. Excessively high or low particle distribution can affect the powder flowability at the powder bed, potentially leading to porosity defects during the L-PBF process [40]. To investigate the powder quality further, the particle distribution was characterised using dynamic image particle analysis to distinguish the mode, mean, median and volume fractions (D_10_, D_50_, and D_90_), which are listed in Table 3.

The Carpenter Additive Cu powder featured a relatively low particle distribution with a D_10_ of 15.1 µm, D_50_ of 30.6 µm and D_90_ of 44.3 µm. As such, Carpenter Additive Cu D_10_ was 26% lower than the next lowest D_10,_ highlighting the finer particle distribution as shown in Figure 2a. All other feedstock powders had relatively similar particle distributions, with a maximum of 10% variation between the lowest and highest D_10_ results. EOS CuCP featured a D_10_ of 19.0 µm, D_50_ of 35.2 µm and D_90_ of 48.5 µm. EOS Cu featured a D_10_ of 21.0 µm, D_50_ of 36.4 µm and D_90_ of 49.4 µm. CuAg10% featured a D_10_ of 19.0 µm, D_50_ of 34.9 µm and D_90_ of 49.6 µm. CuAg20% featured a D_10_ of 19.0 µm, D_50_ of 34.2 µm and D_90_ of 49.0 µm. Legor Ag featured a D_10_ of 20.4 µm, D_50_ of 31.5 µm and D_90_ of 52.0 µm. Therefore, the powder feedstock characteristics shown in Figure 2 and Table 3 suggest that all feedstock powders investigated in this study have desirable spherical morphology and particle distributions required for the L-PBF process to enhance both the packing density and powder flowability [36], and therefore are suitable for L-PBF processing.

### 3.2. L-PBF Processing and Associated Variables 

All IEC electrical conductivity geometry samples were fabricated utilising an EOS M290 L-PBF system with standard 400 W infrared laser, and the parameters as described in Table 2. Figure 3 displays two as-built samples on Cu build platforms, where Figure 3a is fabricated from Carpenter Additive Cu feedstock and Figure 3b is manufactured from Legor Ag feedstock.

### 3.3. L-PBF High-Purity Copper 

Considering the challenges highlighted in the literature [25,26,27,49] regarding the L-PBF processing of high-purity Cu, the initial investigation evaluated the feasibility of L-PBF processing Cu using a 400 W laser system. Three high-purity (>99%) Cu variants listed in Table 4 were utilised for sample manufacture and then characterised for their electrical performance. The process parameters used are listed in Table 2, with single laser exposure and soft recoating strategies remaining constant to ensure study variants were limited to Cu feedstocks purity and PSD characteristics. 

Figure 4a displays the electrical conductivity of the varying purity Cu feedstock EOS Cu (>99%), Carpenter Additive Cu (>99.6%) and EOS CuCP (>99.98%), respectively. The highest electrical conductivity performance was seen by Carpenter Additive Cu, with Cu purity composition between the two EOS variants. Furthermore, EOS Cu with 0.98% lower Cu purity than EOS CuCP displayed relatively similar electrical performance, with a variation of 0.7% in comparison with 14% for Carpenter Additive Cu. CT was utilised to identify any potential contribution from material density that could have contributed to the electrical performance. As such, Figure 5a–c displays CT data for each IEC sample EOS Cu, Carpenter Additive Cu and EOS CuCP, respectively. For all samples analysed, the distribution of the pores along the sample length were consistent, confirming the homogeneity of the process; however, clear differences between sample densities can be seen, with the Carpenter Additive Cu porosity (Figure 5b) being lower than both EOS Cu (Figure 5a) and EOS CuCP (Figure 5c).

Figure 4b displays the relative densities of 73%, 75% and 88% for EOS Cu, EOS CuCP and Carpenter Additive Cu, respectively. The electrical conductivity results (Figure 4a) correlate directly with relative densities, confirming that for L-PBF high-purity Cu electrical conductivity performance is directly related to component density. The data show that when processing the industry-supplied Cu variants with identical L-PBF processing parameters, small enhancements in Cu purity are negated by the negative effects in resultant density. Since Carpenter Additive Cu (with purity composition between that of EOS Cu and EOS CuCP) shows the highest electrical performance, it is evident that the purity differences between variants cannot be attributed to the lower density. Carpenter Additive Cu feedstock particle distribution induced notable increases in density, giving rise to enhanced electrical performance regardless of Cu purity in this case. 

### 3.4. High-Purity Cu, Ag and Cu–Ag Alloys 

The electrical performance of 400 W L-PBF-processed Cu, Ag and Cu–Ag materials and alloys have yet to be reported in the literature. However, the electrical performance of Cu–Ag alloys, have been reported for material fabricated by multilayer, spark plasma sintering [50,51] and cold drawing [52]. In multi-layered, spark-plasma-sintered and cold-drawn Cu–Ag samples, an increase in Cu–Ag grain boundaries was shown to act as electron scattering sites, in turn negatively effecting conduction electrons [51] and electrical performance. Additionally, increased Cu–Ag interfaces can create high dislocation density, in turn reducing the conductive properties of the material [53]. Ag addition in Cu–Ag alloys can improve material strength due to increased Cu–Ag phase boundaries, creating strong binding effects and hindering dislocation movements [21]. Contrastingly, the same mechanisms that enhance material strength can negatively affect electrical performance. Accordingly, L-PBF Cu, Ag and two Cu–Ag alloys, with compositions listed in Table 5, were investigated to evaluate their comparative density and electrical performance.

Figure 6 displays the CT data for IEC samples fabricated using EOS Cu, Cu–Ag10%, Cu–Ag20% and Legor Ag, respectively. Ag addition to Cu has been shown to reduce L-PBF Cu–Ag pore content in previous studies [21], and the X-ray visuals for the Cu–Ag10%, Cu–Ag20% and Legor Ag samples confirmed this, displaying smaller pore defect size when compared with the EOS Cu variant (Figure 6a). As with high-purity Cu, porosity distribution along the length of the sample was consistent, confirming the homogeneity of the process. Cu–Ag10%, Cu–Ag20% and Legor Ag showed smaller pore defect size in comparison with EOS Cu. CT data for Cu–Ag10 (Figure 6b) and Cu–Ag20 (Figure 6c) also indicate consistent and evenly dispersed Cu–Ag interfaces throughout the samples identified as high-density (white spots).

Figure 7 displays EOS Cu, Cu–Ag10%, Cu–Ag20% and Legor Ag electrical performance (Figure 7a) and relative densities (Figure 7b). No corelation was observed between the electrical performance of Cu–Ag alloys and sample density, which is due to the Cu–Ag interface impeding conduction electrons through scattering and dislocation density. Electrical performance variations between EOS Cu and both Cu–Ag alloys was 0.9% and 0.5% IACS, while density variations were 14.1% and 11.6%, respectively. Additionally, increased Ag saw both negative and positive variations in comparison with Cu within 1%. Cu–Ag10% with highest reported density had the lowest electrical performance of all materials, being 58.1% IACS. Although Legor Ag featured similar relative density to both Cu–Ag alloys, the electrical performance was higher by 6.9% and 8.3% due to the lack of Cu–Ag interfaces negatively effecting electron conduction.

#### Powder Recoating and Laser Exposure

L-PBF is a micron-scale layer-by-layer AM process where components are fabricated by the SLM of powder feedstock layer on previously melted powder layers. The powder feedstock is usually transported from a dispensing hopper or powder bed. Therefore, the powder dispensing process itself can influence the resultant dispensed layer characteristics and final component density and properties [54]. Powder rheology and motion dynamics from dispenser to build platform can be critical [55], and therefore recoating speed, dosing factor and recoating technique can all effect powder layer characteristics [56].

For instance, using a soft (brush) blade to dispense powder offers advantages for filigree structures that may be affected by hard (steel/ceramic) blade impact. However, hard blade and roller powder dispensing techniques result in a higher powder packing density and therefore denser fabricated components [54,57]. This is due to compressive/shear mechanisms and larger compact regions positively affecting layer packing density and laser and material interactions such as absorptivity and heat transfer [58]. Additionally, previous studies have shown that increasing Cu powder packing density leads to higher component density [59]. 

However, L-PBF processing of Cu still offers significant challenges due to insufficient keyhole melting [26] and relatively low densities being reported [27], particularly using standard infrared lasers (<500 W). As such, optimising L-PBF processing parameters, including single- and double-laser (remelt) exposure scanning strategies (as shown in Figure 8), could improve Cu processing. Accordingly, this investigation evaluates the influence of both soft and hard recoating blades on the resulting density and electrical conductivity of IEC samples being fabricated and explores the effects of laser exposure duration on density and the resultant electrical performance of L-PBF-processed Cu. Four IEC samples were fabricated using Carpenter Additive Cu feedstock, where soft and hard blade powder recoating and single- and double-exposure strategies (as listed in Table 6) were employed.

Figure 9 summarises the IACS electrical conductivity performance for Carpenter Additive Cu samples #1, #2, #3 and #4. Hard blade recoating resulted in higher electrical performance for both single- and double-exposure strategies, leading to improvements of 1.7% and 1.1%, respectively. The single-laser exposure strategies were shown to improve electrical conductivity performance, with single-exposure samples outperforming both double-exposure samples by 1% and 0.3%.

Figure 10 displays the CT data for the highest (#3) and lowest performing (#2) IEC samples. Sample #3 with single-laser exposure and hard recoat parameters resulted in a relative density of 88.15% higher than that of #2, which showed a density of 85.37%. While difficult to separate visually, the volumetric data show that the hard recoating methods and single-laser exposure strategy result in superior sample density, subsequently translating to higher electrical conductivity. 

### 3.5. Surface Roughness 

For AM surface roughness analysis, the arithmetic average roughness value (Ra) is the generally agreed industry term. L-PBF material feedstock, build setup and processing parameters can all have a significant effect on fabricated component surface roughness. Therefore, the surface roughness of L-PBF parts can to some extent be controlled dependant on material, processing and laser parameters. The as-built L-PBF surface finish is often dictated by unmolten powder particles due to the powder bed thermal process. Generally, the industry practice is to complete post-build abrasive cleaning to remove unmolten powder to create the finished part [60,61]. Accordingly, the as-built surfaces referred to in this study underwent an abrasive cleaning process.

Furthermore, any variation in build orientation or down-skin areas can result in a stair effect. Therefore, Ra values reported are indicative of horizontal or vertical surfaces. For example, following abrasive post-processing, the vertical and horizontal surfaces of common AM metal materials are Ti64 Ra 5–9 µm [62], AlSi10Mg Ra 9–20 µm [61] and TiCP Ra < 10 µm [60], respectively.

Cu reflectivity, heat dissipation and oxidation tendency can attribute to poor surface finish for L-PBF Cu, with previous studies reporting Ra 18–30 μm (pure copper) [63] and Ra 10–16 μm (CuCrZr) [63]. Accordingly, the surface roughness of three Carpenter Additive Cu samples were evaluated using a Keyence VHX-7000 digital microscope, as summarised in Table 7, with representative 3D visualisations displayed in Figure 11. Sample #1 underwent abrasive blasting to remove unmolten powder without electropolishing, referred to as built. Sample 2.1 and 2.2 underwent the same abrasive blasting process followed by electropolishing techniques at different current densities and time durations, as listed in Table 7.

Sample 2.1 was processed using an average current density used for electropolishing similar copper grades. Although sample 2.1 showed a visual increase in lustre, a higher current density was required for sufficient material removal. Therefore, sample 2.2 was subject to double the average current density (at 0.8 a/cm^2^) and received two separate 10 min immersions with a break in between to ensure no burning/process errors. Additional processing would eventually result in preferential dissolution of the corners and any sharp edge, and therefore processing was capped at 10 min per immersion to avoid compromising the overall geometry.

Although sample 2.1 showed a higher lustre, a 10 min immersion at 0.4 a/cm^2^ had no effect on surface roughness, leading to almost identical Ra values to the as-built sample (#1), being 6.40 µm and 6.42 µm, respectively. Sample 2.2, being exposed to two 10 min immersions at 0.8 a/cm^2,^ decreased the surface roughness to 2.78 µm, highlighting the potential of the electropolishing process for improved surface finish for L-PBF Cu components. It was also found that electropolishing regimes had limited effect on electrical conductivity (as displayed in Figure 12) due to the performance being largely driven by sample density. As such, variations from the as-built condition were 0.3% for sample 2.1 (Figure 12a) and 0.9% for sample 2.2, as shown in Figure 12b.

### 3.6. Electrical Performance Summary

The L-PBF processing of high-purity Cu, Ag and Cu–Ag alloys has been reported here for the first time, with material composition (Cu, Ag and Cu–Ag), powder dispensing, laser exposure and surface roughness all analysed as input variables. So far, the discussion has focused on each variable, however, Figure 13 brings together the electrical conductivity and relative sample densities for all Cu purities, Ag, Cu–Ag alloys and other variables to allow for holistic analysis and to identify performance trends. 

Overall, Carpenter Additive Cu consistently outperformed all other Cu variants, Cu–Ag alloys and Ag material for electrical performance, regardless of Cu purity or Ag content. Additionally, for Cu samples, the density correlated directly to the electrical performance, with Carpenter Additive Cu sample densities being consistently higher in comparison with EOS Cu and EOS CuCP. This is explained when evaluating the feedstock particle distribution displayed in Figure 2 and Table 3. Carpenter Additive Cu feedstock exhibited a lower PSD than other variants, with a D_10_ that is 26% lower than the next lowest variant. The lower PSD displayed includes a higher proportion of particles below 20 µm, resulting in greater layer packing density during the L-PBF process. Consequently, the increased layer packing density increased overall sample densities, leading to enhanced electrical performance. Additionally, hard blade recoating, which is known to improve layer thickness packing density due to compressive and shear mechanisms, also improved electrical performance further. Contrastingly, density and electrical performance for Cu–Ag alloys did not follow the same trend, with increased density having no obvious correlation with Cu–Ag electrical performance. This is explained by electron dispersion and scattering mechanisms created by Cu–Ag interfaces and dislocation density sites, as discussed in Section 3.4. 

### 3.7. Prospects for L-PBF Cu and Ag

Overcoming L-PBF Cu and Ag processing challenges could allow for the development of high-performance Cu- and Ag-based materials and alloys [17], suitable for electrical windings in motors [64], rocket launchers and heat sinks [65]. As such, L-PBF-processed conductive architecture will complement developments in CAD and simulation techniques, design tools [66,67], custom materials [68,69] and AM technologies [70,71], offering superior performance components for space, green energy and aerospace and automotive sectors [72]. L-PBF AM will therefore allow the fabrication of custom shape and custom volume conductor designs with desirable incorporated features such as cooling channels for electrical machine thermal management. Consequently, it is expected that future designers and engineers will have application-specific material and manufacturing capabilities to conceive cost-effective and superior performance application-specific solutions. 

#### Cu Coil Winding and TPMS Heat Sink Proof of Concept

Finally, coil winding geometries and Cu L-PBF process parameters were used to assess the feasibility of L-PBF fabrication of pure Cu-shaped profile coil windings. Figure 14 shows proof-of-concept coil windings fabricated from high-purity Cu using a 400 W standard laser L-PBF system. The coil features a Triply Periodic Minimal Surface (TPMS) architecture that acts as a heat exchanger for the coil winding design to demonstrate the potential for incorporated thermal management. 

## 4. Conclusions 

In this work, high-purity copper (Cu), silver (Ag) and copper–silver (Cu–Ag) alloys were additively manufactured using a standard 400 W Laser Powder Bed Fusion (L-PBF) system, and their resultant densities and electrical conductivity performances were evaluated. Six highly reflective and thermally conductive Cu and Ag material variants were comparatively investigated focusing on material purity, Ag content in Cu–Ag alloys, recoating, laser exposure and electropolishing. The results show a link between Cu feedstock characteristics and the resulting electrical conductivity of the L-PBF processed samples. Looking at the influence of Cu purity on electrical performance, no corelation was found, with the highest purity Cu (>99.98%) performing similar to (59.7% and 59% IACS) the lowest purity Cu (>99%). For Cu, higher layer packing density dictated by lower feedstock PSD had a significant positive effect on sample density and electrical performance. When it comes to Cu–Ag, the relative densities and electrical performance were found to be unrelated. This is primarily due to electrical performance being restricted by Cu–Ag interfaces negatively effecting electron conduction through dislocation density and scattering mechanisms. However, it was found that L-PBF process parameters utilised for sample fabrication resulted in denser Ag in comparison with Cu, which consequently resulted in comparatively higher electrical performance. Looking at the impact of powder recoating and laser exposure on L-PBF Cu processing, hard blade recoating and single-laser exposure strategies showed improvements in density, with increases of 2.8% and 2% IACS electrical conductivity for Cu when compared with soft blade recoating and double-laser exposure strategies. Lastly, electropolishing was found to improve the Cu surface roughness from Ra of 6.42 µm to 2.78 µm. Nevertheless, current densities and immersion durations twice that of electropolishing similar copper grades manufactured by other methods are required for L-PBF-processed variants. Overall, the results show that contrary to recent literature and commonly agreed understanding, L-PBF processing of high-purity Cu, Ag and Cu–Ag is feasible using a standard laser 400 W L-PBF AM system. Additionally, further improvements in component density and electrical performance could be achieved through the optimisation of powder feedstock PSD and L-PBF process parameters specifically for Cu 400 W processing. 

## Figures and Tables

**Figure 1 materials-15-07563-f001:**
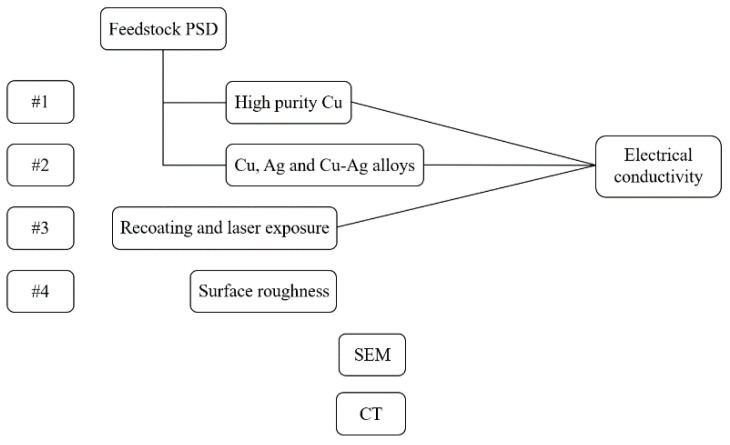
Schematic illustration of investigations conducted.

**Figure 2 materials-15-07563-f002:**
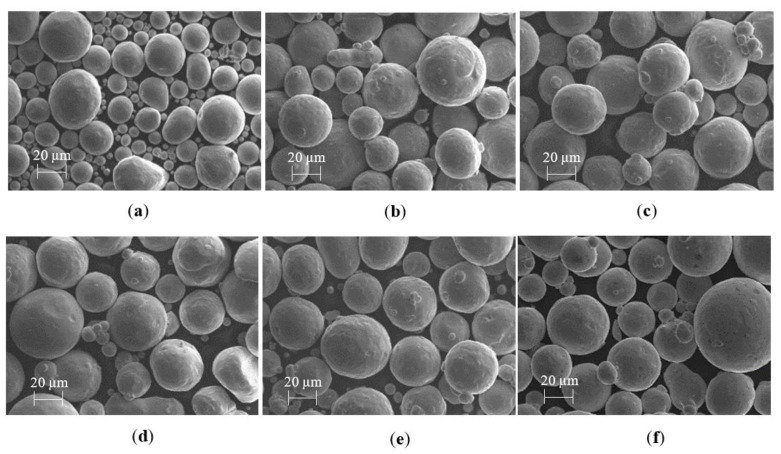
Scanning Electron Microscopy (SEM) data showing particle morphology for (**a**) Carpenter Additive Cu, (**b**) EOS CuCP, (**c**) EOS Cu, (**d**) CuAg10%, (**e**) CuAg20% and (**f**) Legor Ag.

**Figure 3 materials-15-07563-f003:**
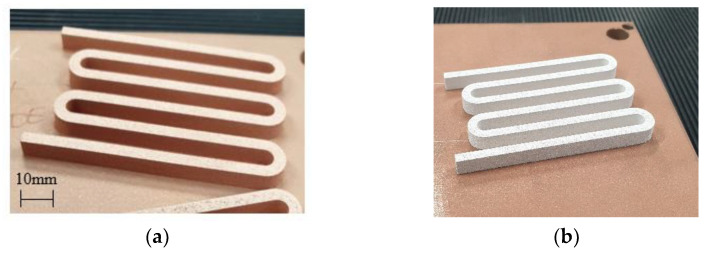
High-purity copper and silver IEC electrical conductivity geometry samples as built on copper substrates, where (**a**) is Carpenter Additive Cu and (**b**) is Legor Ag.

**Figure 4 materials-15-07563-f004:**
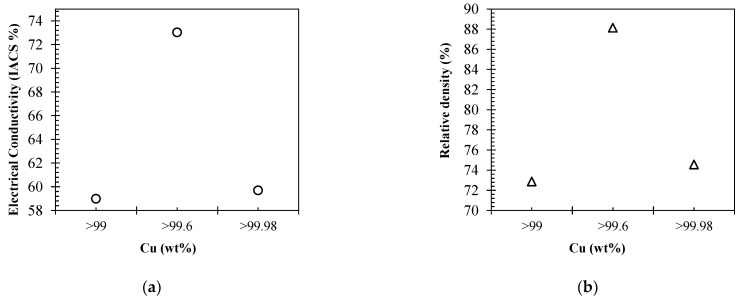
IACS electrical conductivity performance (**a**) and XCT relative density (**b**) for EOS Cu (>99%), Carpenter Additive Cu (>99.6%) and EOS CuCP (>99.98%) high-purity copper feedstock.

**Figure 5 materials-15-07563-f005:**
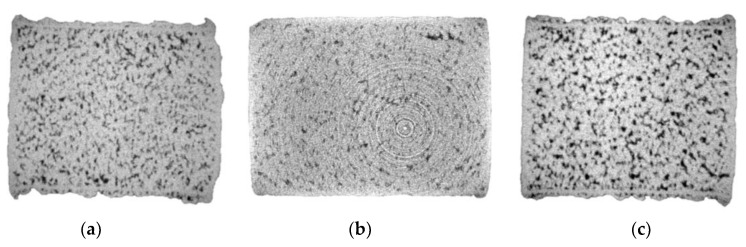
Computed Tomography data for high-purity Cu, where (**a**) is EOS Cu, (**b**) is Carpenter Additive Cu and (**c**) is EOS CuCP.

**Figure 6 materials-15-07563-f006:**
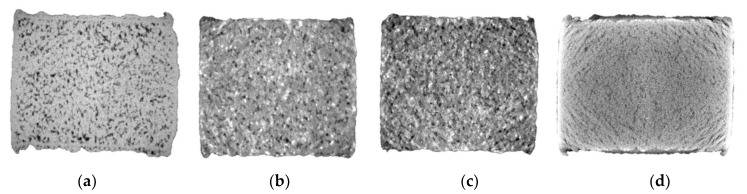
Computed Tomography data for high-purity Cu, Ag and Cu–Ag alloys, where (**a**) is EOS Cu, (**b**) is Cu–Ag10%, (**c**) is Cu–Ag20% and (**d**) is Legor Ag.

**Figure 7 materials-15-07563-f007:**
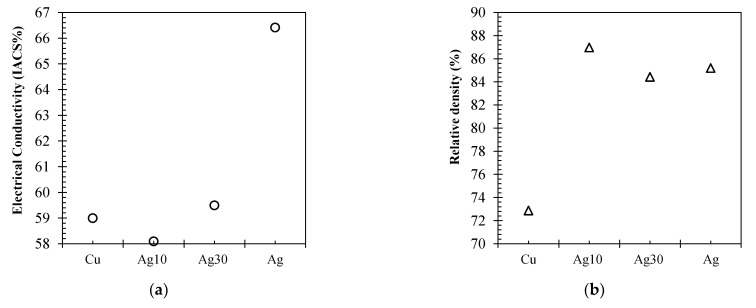
IACS electrical conductivity performance (**a**) and XCT relative density (**b**) for EOS Cu (>99%), CuAg10, CuAg20 and Legor Ag high-purity feedstocks.

**Figure 8 materials-15-07563-f008:**
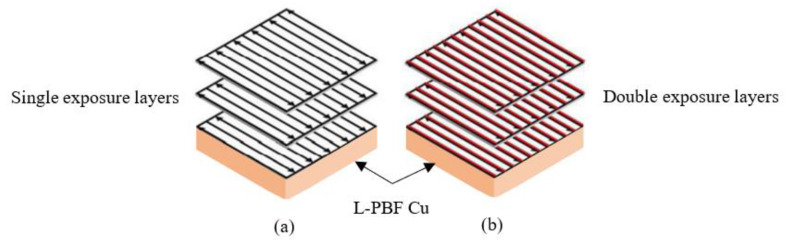
Schematic of laser scanning strategies, where (**a**) is single exposure and (**b**) is double exposure.

**Figure 9 materials-15-07563-f009:**
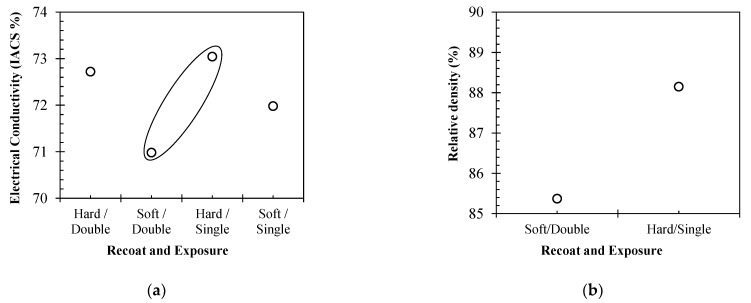
(**a**) IACS electrical conductivity performance for powder recoating and laser exposure parameters, (**b**) Relative densities obtained for powder recoating and laser exposure parameters.

**Figure 10 materials-15-07563-f010:**
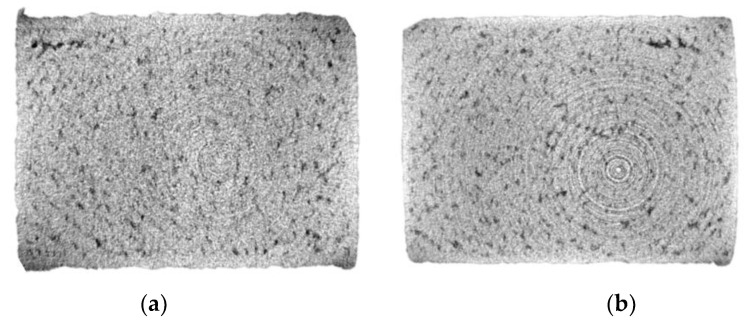
Computed Tomography data of high-purity copper, where (**a**) is Carpenter Additive Cu #2 (porosity 14.63%) and (**b**) is Carpenter Additive Cu #3 (porosity 11.85%).

**Figure 11 materials-15-07563-f011:**
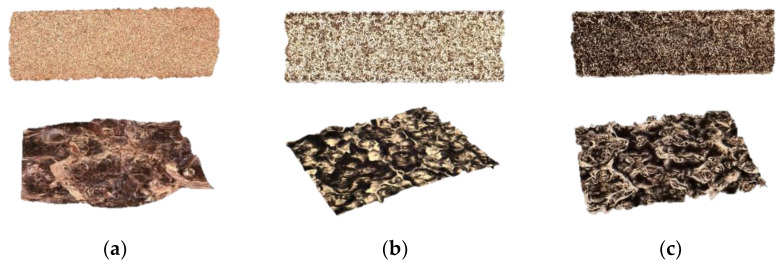
Surface roughness 3D visualisation of as-built and electropolished samples, where (**a**) is as-built, (**b**) is 0.4 a/cm^2^ current density and (**c**) is 0.8 a/cm^2^ current density.

**Figure 12 materials-15-07563-f012:**
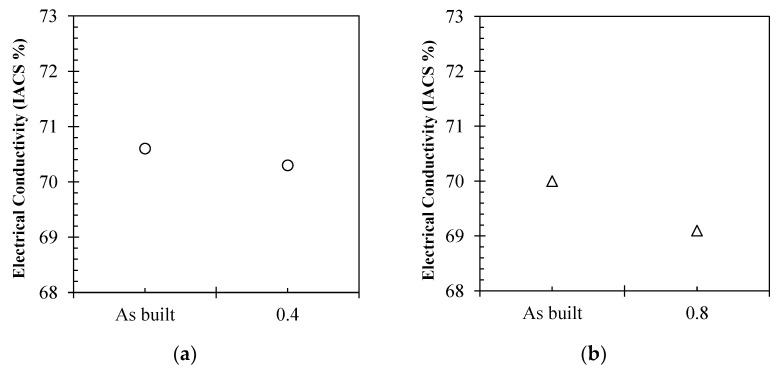
Electrical conductivity performance of (**a**) sample 2.1 as built and following 0.4 a/cm^2^ current density electropolishing regime and (**b**) sample 2.2 as built and following 0.8 a/cm^2^ current density electropolishing regime.

**Figure 13 materials-15-07563-f013:**
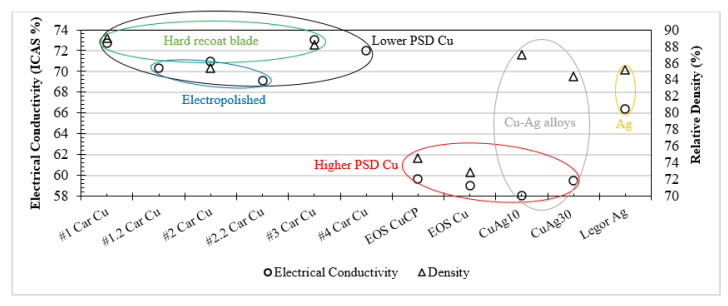
Summary of electrical performance and relative density results for all material compositions and process variables considered.

**Figure 14 materials-15-07563-f014:**
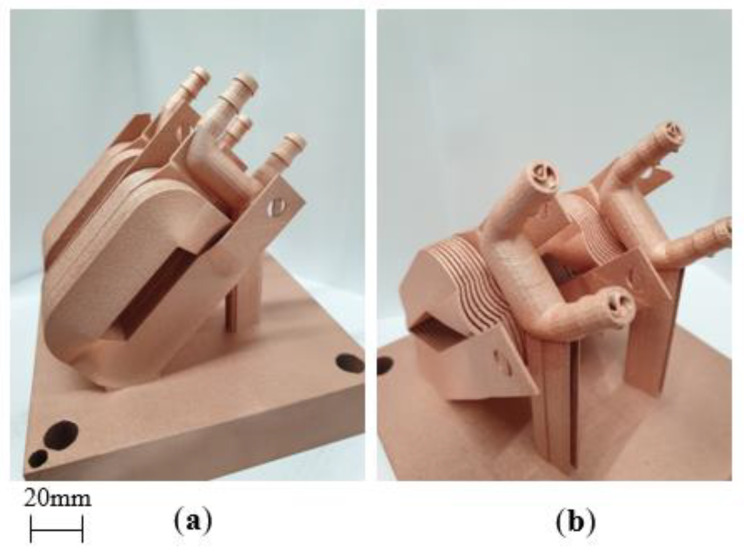
Examples of pure Cu coil winding and incorporated Triply Periodic Minimal Surface heat exchanger samples, where (**a**) displays the coil winding structure and (**b**) displays heat exchanger with internal TPMS structures.

**Table 1 materials-15-07563-t001:** Material suppliers and element weight (%) of Cu and Ag feedstock.

Material	Element	Weight (%)
Carpenter Additive Cu	Cu	>99.6
EOS CuCP	Cu	>99.98
EOS Cu	Cu	>99
Copper–silver 10%	Cu–Ag10	~10% Ag
Copper–silver 20%	Cu–Ag20	~20% Ag
Legor Silver	Ag	100

**Table 2 materials-15-07563-t002:** L-PBF process parameters used for sample fabrication.

Laser Power	Scan Speed	Hatch Distance	Layer Thickness
370 W	400 mm/s	0.14 mm	30 µm

**Table 3 materials-15-07563-t003:** High-purity Cu and Ag powder particle volume distribution.

Material	D_10_ (µm)	D_50_ (µm)	D_90_ (µm)
Carpenter Additive Cu	15.1	30.6	44.3
EOS CuCP	19.4	35.2	48.5
EOS Cu	21.0	36.4	49.4
CuAg10%	19.0	34.9	49.6
CuAg20%	19.0	34.2	49.0
Legor Ag	20.4	31.5	52.0

**Table 4 materials-15-07563-t004:** High-purity Cu variants and supplier-specified purity.

Material	Weight (%)
EOS Cu	>99.0
Carpenter Additive Cu	>99.6
EOS CuCP	>99.98

**Table 5 materials-15-07563-t005:** Material composition of silver addition study.

Material	Element	Weight (%)
EOS Cu	Cu	>99
EOS Cu–Legor Ag	Cu–Ag	~Ag10
EOS Cu–Legor Ag	Cu–Ag	~Ag20
Legor Ag	Ag	100

**Table 6 materials-15-07563-t006:** Recoat and exposure study.

Identifier	Material	Recoat	Exposure
#1	Carpenter Additive Cu	Hard	Double
#2	Carpenter Additive Cu	Soft	Double
#3	Carpenter Additive Cu	Hard	Single
#4	Carpenter Additive Cu	Soft	Single

**Table 7 materials-15-07563-t007:** Surface roughness of L-PBF as built and electropolished copper.

Identifier	Material	Duration (m)	Current Density	Ra (µm)
#1	Carpenter Additive Cu	0	0	6.40
#2.1	Carpenter Additive Cu	10	0.4 a/cm^2^	6.42
#2.2	Carpenter Additive Cu	20	0.8 a/cm^2^	2.78

## Data Availability

The data that support the findings of this study are available from the corresponding author upon reasonable request.

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
