# Peer review of "Electrical Conductivity of Additively Manufactured Copper and Silver for Electrical Winding Applications"

_materials, 2022, doi:10.3390/ma15217563_

Round 1

Reviewer 1 Report

Article focuses Electrical conductivity of additively manufactured copper and silver for electrical winding applications. Following comments need to address before acceptance.

1.      Reduce the section 2 (Materials and Methods). You need to be little concise

2.      Is L-PBF parameters same for all three AM printed samples?

3.      Merge figure 12 (a) and (b) graph

4.      Merge figure 9 (a) and (b) graph

5.      Remove the horizontal lines from the graphs fig. (5, 7, 9 and 12)

Reviewer 2 Report

The paper presents a study about electrical conductivity of copper and silver for electrical windings. Authors say, that power dense electrical devices are critical in driving the next generation of green energy technologies for many industries. One of the primary requirements to enable this is the fabrication of compact custom windings with optimized materials and geometries. Electrical machine windings rely on highly electrically conductive materials. Authors investigated L-PBF processing of high purity Cu, Ag and Cu-Ag alloys and the resultant electrical conductivity performance.

Thank you very much for the paper about electrical conductivity of copper and silver, which is used in case of electrical windings of devices I put some comments and suggestions.

1. In Introduction chapter, authors present fundamental information about electrical machines. They put attention on high power machines, indicating parameters, which play important role in order to minimalizing the costs of application of mentioned devices. Authors are focused on ration power – weight, which should be possible reduced as much as possible.

2. Next chapter presents used materials and methods, which was used in investigations. Authors describe laser powder bed fusion, mentioned powder characterization, and L-PBF parameters. They explain annealing process, surface roughness, density and electrical conductivity.

3. Title of the article refers to electrical conductivity. In the same time, subchapter, which describes the conductivity, is the smallest chapter in chapter 2. I propose to extend the subchapter 2.7. This subchapter should present fundamental information and physical function of electrical conductivity.

4. Next chapter escribes obtained results. Authors present studied powder and L-PBF processing.

5. Fig. 5.a. presents relationships of electrical conductivity as a function of Cu weight. There is no logical relation. Please explain it in the text. The same comment is for figure 5.b. Please complete it using some explanation. The same conclusion is for Fig. 9.a.

6. Subchapters 2.2. and 3.1. have the same name – Powder Characterization. I think it is not correct. Please change it and propose some other name for one of the subchapters.

Reviewer 3 Report

1. It is a strange way to write in two paragraphs, and there are too many words in Abstract.

2. It is not clear why a hard blade can achieve higher sample density. It is not necessarily that a hard blade can obtain a higher packing density of the powder layer. Similarly, many of the assertions in the article are not supported by evidence.

3. This study does not clearly explain why the finer Carpenter Additive Cu powder can obtain higher density samples. Different particle size distribution of powder will cause significant difference in laser energy absorption. The author needs to pay attention to this issue.

4. It does not mean that the finer the powder, the higher the packing density of the powder layer. Therefore, the author needs to give the packing density of various powders. Moreover, it does not mean that the higher the powder packing density is, the greater the density of the sample will be.

5. To discuss the forming density of the sample, it is necessary to consider the interaction between laser and powder. That is, the energy absorption of the powder layer to the laser and the dynamic stability of the molten pool.

Round 2

Reviewer 3 Report

It can be accepted in present form.